# Developing New Anti-Tuberculosis Vaccines: Focus on Adjuvants

**DOI:** 10.3390/cells10010078

**Published:** 2021-01-05

**Authors:** Ana Rita Franco, Francesco Peri

**Affiliations:** Department of Biotechnology and Biosciences, University of Milano-Bicocca, 20126 Milano, Italy; anarita.adelinofranco@unimib.it

**Keywords:** vaccines, adjuvants, infectious diseases, tuberculosis

## Abstract

Tuberculosis (TB) is an infectious disease caused by *Mycobacterium tuberculosis* (Mtb) that sits in the top 10 leading causes of death in the world today and is the current leading cause of death among infectious diseases. Although there is a licensed vaccine against TB, the *Mycobacterium bovis* bacilli Calmette–Guérin (BCG) vaccine, it has several limitations, namely its high variability of efficacy in the population and low protection against pulmonary tuberculosis. New vaccines for TB are needed. The World Health Organization (WHO) considers the development and implementation of new TB vaccines to be a priority. Subunit vaccines are promising candidates since they can overcome safety concerns and optimize antigen targeting. Nevertheless, these vaccines need adjuvants in their formulation in order to increase immunogenicity, decrease the needed antigen dose, ensure a targeted delivery and optimize the antigens delivery and interaction with the immune cells. This review aims to focus on adjuvants being used in new formulations of TB vaccines, namely candidates already in clinical trials and others in preclinical development. Although no correlates of protection are defined, most research lines in the field of TB vaccination focus on T-helper 1 (Th1) type of response, namely polyfunctional CD4+ cells expressing simultaneously IFN-γ, TNF-α, and IL-2 cytokines, and also Th17 responses. Accordingly, most of the adjuvants reviewed here are able to promote such responses. In the future, it might be advantageous to consider a wider array of immune parameters to better understand the role of adjuvants in TB immunity and establish correlates of protection.

## 1. Introduction

Tuberculosis (TB) is an infectious disease caused by *Mycobacterium tuberculosis* (Mtb), that has been included in the top 10 leading causes of death in the world today and is the current leading cause of death among infectious diseases [1,2]. This devastating title came at the expense of 1.4 million lives in 2019 alone [3]. According to the World Health Organization (WHO), a quarter of the world population is infected with Mtb [3] and most of these cases are related to poverty and located in low and middle-income countries [2,3].

Although there is a licensed vaccine against TB, the *Mycobacterium bovis* bacilli Calmette–Guérin (BCG) vaccine, it has several limitations, namely its high variability of efficacy in the population and low protection against pulmonary tuberculosis [4]. This explains the number of Mtb infections worldwide, despite the existence of a vaccine and its availability.

Treatment for TB normally consists of a long antibiotic regimen that decreases the patients’ compliance [5]. Nonetheless, the efficacy of this treatment is now compromised by the emergence of multidrug-resistant *Mycobacterium tuberculosis* bacteria against which there is only 50% chance of cure with the available drug treatment [6]. The WHO has set the goal to reduce TB morbidity by 90% and mortality by 95% by 2035 [6].

While drug research and diagnostics need to evolve in order to achieve control over Mtb infections, the discovery of a new and effective vaccine takes a central stage for its cost-effectiveness and ability to prevent or help with the treatment of TB, even of multidrug resistant strains [4,6,7]. By using a vaccine, it is possible to decrease both the transmission of the pathogen and the use of antibiotics to treat it, which means that the vaccine controls the disease and overcomes the antimicrobial resistance (AMR) [7].

Currently, in the TB vaccine pipeline, there are different types of vaccines, including live attenuated, inactive and subunit vaccine candidates [4,8]. Subunit vaccines are not the most advanced at the moment, since the current phase III clinical trials only comprise whole-cell vaccines, however they are promising candidates [9]. This type of vaccines overcomes some of the safety concerns associated with live attenuated or inactive vaccines [10]. Furthermore, as they are designed using specific antigens, they can be targeting different aspects of the infection, optimized as the research progresses and provide additional immune responses for more complex pathogens, such as Mtb [8,11].

While being very safe, subunit vaccines cannot rely solely on the antigen to achieve a desired effect as they are generally poorly immunogenic [10]. Adjuvants are needed in their formulation in order to increase immunogenicity, decrease the antigen dose, ensure a targeted delivery and optimize the antigens interaction with the immune cells [10,11]. Undeniably, adjuvants are an important part of new subunit vaccines design and in general, adjuvants identification, design and characterization is essential to modern vaccines formulation [12]. Additionally, understanding how adjuvants work plays a determinant role in vaccine development and can be the key factor to make or break the vaccine’s chances to succeed [12].

This review focuses on adjuvants being used in new formulations of TB vaccines, namely candidates already in clinical trials and others in an initial research state, the immune responses that they elicit, and the known outcomes so far.

## 2. The BCG Problem and the Design of New TB Vaccines

BCG is a live attenuated vaccine produced using *Mycobacterium bovis* [13] that has been in use for over 100 years [9,14]. It is an inexpensive, widely available vaccine that is administrated to more than 90% of children in endemic countries [14,15]. The outcomes of this vaccine are very positive when it comes to prevention of childhood meningeal or miliary tuberculosis and overall decrease in child mortality, but the same vaccine fails to protect against adult pulmonary tuberculosis in a homogenous manner in the world population, with highly variable protection ranging from 0% to 80% [7,10,15].

While BCG vaccine has many downsides when pulmonary adult tuberculosis is being analysed, it is still an excellent vaccine that prevents child tuberculosis with low costs and. Therefore, BCG boosting lines of research should also be considered as a good starting point for TB vaccine development [16].

WHO considers the development and implementation of new TB vaccines to be a priority [16]. Accordingly, in 2017, the organization proposed a guideline for the development of these vaccines called “WHO preferred product characteristics” and presented to experts from different branches of the industry, such as scientists, funding agencies and regulators [16].

Given the complexity of the disease and to target efficiently the different states of infection, latent or active, TB vaccination should provide several layers of protection, namely by preventing initial infection, reactivation of infection or progression into active disease [14,16]. Moreover, the vaccine should also be suited and safe for administration in immunocompromised patients like HIV positive individuals [14]. While the WHO also describes the need for research on a new newborn/infants TB vaccine [15], this review mainly focuses on the efforts to develop a successful vaccine for pulmonary tuberculosis in adults and adolescents.

A summary of the vaccine’s preferred characteristics, for a target population of adolescents and adults, is presented in the Table 1.

## 3. Understanding the Adjuvant’s Immune Role by Understanding TB Immunity

After inhalation of infected aerosols, the phagocytosis of the Mtb pathogen by the alveolar macrophages takes place. The block of the lysosome-phagosome fusion leads to the survival of the bacterium within the macrophages with activation of a cell-mediated immune response, such as CD4+ and CD8+ T lymphocytes [17]. From there, the immune response leads to a very important and characteristic cell-mediated consequence which is the formation of a granuloma [17]. This structure composed by macrophages, lymphocytes, stem cells and epithelial cells has the ability to control bacterial replication and induce a latent stage of the disease [17]. Understanding this immune response and the mediators involved is essential for the development of a vaccine and for choosing an appropriate adjuvant. The main problem with a TB vaccine is that the host immune responses towards Mtb are not yet fully understood and, as said before, there is no defined correlate of protection [18].

Most research lines in the field of TB vaccination focus on T-helper 1 (Th1) type of response, namely polyfunctional CD4+ cells expressing simultaneously IFN-γ, TNF-α, and IL-2 cytokines [13,19,20]. While it seems that investigators are moving towards the discovery and establishment of a correlate of protection, the findings are not clear. Looking at the pathogen’s behaviour to understand if these cells are indicative of protection, it is possible to see some ambiguous results since CD4+ cells are related to protection against disease and with immune responses to successful treatments but a high expression of these cells are also related to an increase in the bacterial load during active infection [19]. Moreover, most of the studies already concentrate in finding these cells and thus it is possible that there is some other immune correlate that is not being investigated [19].

Nevertheless, studies on mice and on patients have shown a clear importance of polyfunctional Th1 cells in TB protection and, thus, Th1 is still a desired response in new vaccine candidates [12,19]. In fact, as will be shown in this review, most of these new formulations show protective results and a clear Th1 response.

Th17 cells have also been described as important for immunity against Mtb and they are an immune marker used in some studies [12,17]. It has been previously described, in appropriate mice models, that IL-17 does not play an important role on the early control of the bacteria in the lung, upon infection with Mtb. However, Th17 cells are important for neutrophil recruitment and they are induced by the Mtb infection [20]. The same authors describe that IL-17-producing T-cells may be important for vaccine-mediated protection due to their probable ability to populate the lung and other tissues and, upon infection, start a signal that leads to bacterial load control [20].

In the selection of adjuvants, it is important to categorize the type of response that they are capable to promote, especially when combined with antigens specific for Mtb [18]. It is clear that CD4 T cells are crucial for protection as well as the production of IFN-γ and TNF-α although not enough [18,21]. While other immune mediators are not yet established as essential, their role in Mtb immunity is worth exploring and adjuvant’s associated response in new vaccine candidates is an excellent resource for new information regarding these immune responses.

## 4. Adjuvants in New TB Vaccine Candidates

This section aims to give insights on adjuvants that are being used in innovative TB vaccine candidates. The available data on these new formulations are reported and discussed. The immune response that they trigger will be explored by analysing the available data, which can potentially contribute to the discussion on vaccine formulation. A brief overview of the mechanism of action of the adjuvants described in this review is presented in Figure 1.

### 4.1. Adjuvants in TB Vaccines Currently in Clinical Stage of Development

#### 4.1.1. IC31

IC31 is a vaccine adjuvant constituted by antimicrobial peptide KLK and oligodeoxynucleotide (ODN) 1a (ODN1a) [22]. The leucine-rich peptide KLK and ODN1a have a synergistic effect in IC31 adjuvant activity.

This adjuvant has a toll-like receptor (TLR) 9 agonist activity that relies on both components [23]. The mechanism of action is related to the ability of TLR9 to be activated by bacterial DNA, namely unmethylated CpG ODN [24], which is the case of ODN1a. In a natural infection with Mtb, this portions of bacterial DNA binds to the TLR9 receptors and promotes macrophage activation and the release of pro-inflammatory cytokines [24].

In a recent report, it has been shown that CpG motifs can be used alone as a TLR9 adjuvant to elicit T-cell response in the mucosa [25]. In this in vivo study in mice, the authors showed that CpG type C, formulated in a liposome in combination with the antigen ESAT-6 and administrated intranasally, was able to reduce significantly the bacterial burden in the lung and that this adjutancy was probably due to type I IFN response by the activation of IFN-α and dendritic cell (DC) activation alongside a IFN-γ Th1 mechanism [25]. This study, along with others cited by the authors, makes an interesting contribution regarding immune responses to TB and vaccine candidates. While the type I IFN response, during a natural infection with Mtb, is often associated with an exacerbation of the disease, in this case, the response was protection [25]. The hypothesis is that, when this pathway is acutely stimulated by a vaccine candidate, it induces a proinflammatory route but when the stimulation is chronical, during active TB disease, the type I IFN has a immunosuppressive effect [25].

In IC31, KLK has the role of delivering the ODN1a to the endosomal TLR9 receptor [23,26]. This peptide is valuable to the formulation since it can translocate into the cells without cell membrane permeabilization, which makes it ideal for ODN1a delivery to the TLR9 receptors and also for antigen presentation [23,27]. Alone, KLK can induce activation of neutrophils and monocytes and promote a Th2 type response [23].

H4:IC31 is a vaccine candidate, which is administrated intramuscularly, where the adjuvant and antigen are kept together by electrostatic interactions, due to the positive charge in IC31 and overall negative charge in H4 [26]. The H4 antigen is actually a fusion protein between Ag85B and TB10.4 Mtb antigens [26,27,28], and its concentration is nine times less than the adjuvants, in most formulations [26].

According to preclinical studies, in a clinical trial report (NCT02066428), it has been shown that IC31 adjuvant responses are importance for protection against TB in the H4:IC31 vaccine candidate [27]. In this dose-escalation phase I trial, the authors determined that the H4 antigen alone did not result in favourable T-cell response while the combination with IC31 resulted in Th1 type response with the proliferation of polyfunctional T-cells that expressed IFN-γ IL-2 and TNF-α for at least 18 weeks [27]. Although this trial was conducted in countries with low TB incidence, Sweden and Finland, the immunological results were similar to another phase I trial conducted in South Africa, a county that is endemic to TB [29]. In a subsequent trial, Nemes and co-workers attempted to compare the protection ability of the H4:IC31 vaccine with BCG revaccination (NCT02075203) [28]. This trial confirmed that the immune responses that the vaccine elicited were CD4+ polyfunctional T-cells, which was in agreement with previous preclinical models and phase I trials [28,30]. Although the immune responses were promising, the clinical trial endpoint was not met with success and the vaccine did not show a desired efficacy in TB prevention [28]. While the trial’s results were disappointing, it had several limitations, namely in the detection of Mtb infection in the participants, which might have influenced the efficacy of the vaccine. Furthermore, while this particular candidate did not thrive, it is clear that the IC31 adjuvant is promising.

While H4:IC31 is a preventive vaccine candidate, the H56:IC31 vaccine is not only preventive [31] but is also being studied as a post-exposure vaccine administered by a intramuscular route. A clinical phase I trial designed to evaluate the immunogenicity and safety of this vaccine in adults with and without Mtb infection (NCT01967134) provided more insights about the IC31 adjuvant. The vaccine candidate was able to induce a specific antigen CD4+ T-cell response, characterized by polyfunctional Th1 type cells, which were in higher frequency in the Mtb infected participants, suggesting that Mtb infection sensitizes the immune system [22].These results were confirmed in a subsequent dose-optimization trial (NCT01865487) [31]. There is currently an ongoing phase II trial studying the ability of H56:IC31 to prevent TB recurrence (NCT03512249). Interestingly, the immune modulation ability of this candidate is also being exploited as a combination therapy with and anti-inflammatory drug in another clinical trial (NCT02503839).

#### 4.1.2. GLA-SE

GLA is the synthetic version of 3-O-desacyl-4′-monophosphoryl lipid A (MPL), with activity as agonist of the TLR4 receptor. GLA-SE adjuvant is GLA formulated in an oil-in-water squalene emulsion [32,33]. During the development of this adjuvant system, the formulation was studied on cells and animals, and it was concluded that although GLA formulated as an aqueous nano-suspension had better results on cell-lines, the in vivo results were very different and GLA-SE showed better immune outcomes [32]. The immune responses that this adjuvant trigger are Th1-biased and rely, from the molecular point of view, on Type I and II interferon responses associated with the production of IL-12 [33].

This adjuvant has been used in many different vaccine candidates not only for TB but also HIV, malaria, leishmaniosis, and other infectious diseases [34]. GLA-SE has been recently used in combination with the antigen ID93 in preclinical and clinical trials [34,35,36,37,38,39]. ID93 is a fusion protein that comprises five Mtb antigens, four of them are associated with virulence and one, Rv1813, with latency [38]. This is important to have an effective vaccine that also targets the latent phase of the infection. Based on extensive preclinical studies, this vaccine candidate has been tested as a prophylactic vaccine in adults vaccinated with BCG (NCT01927159) [38] and non-vaccinated (NCT01599897) [36] and also as a therapeutic vaccine in studies on mice, in combination with antibiotics [37]. In the trial that assessed this candidate as a BCG booster, the ID93:GLA-SE vaccine was administrated intramuscularly to participants that had previously been vaccinated with BCG and that were infected with Mtb or non-infected [38]. The trial was conducted in South Africa which is a country with high-burden of TB and had some limitations, namely the number of participants and the uneven distribution of infected an non-infected patients in the two groups [38]. Nevertheless, it was possible to determine that the vaccine was safe and to draw some preliminary conclusions on the immunogenicity of this vaccine candidate, namely specific IgG production and Th1 response. Furthermore, Mtb infected patients had earlier boosting and a greater T-cell differentiation, compared to the non-infected participants, which implies that the infection primed the response and that the vaccine response can be modulated by the infection itself [38]. Additionally, if a vaccine is successful in inducing immunity in already infected patients it is possible that it can contribute to protection against reactivation of the disease, which is especially relevant in countries like South Africa where a considerable portion of the population has latent TB [38]. In a parallel trial, the vaccine was evaluated as a prophylactic vaccine in non-vaccinated BCG participants and similar to the study described above, this one also was in South Africa and also had some limitations that, in this case, were related to the study design that limited the analysis of the same parameters in the same way in all of the cohorts [36]. Nonetheless, it was possible to observe an increased Th1 response in the group vaccinated with ID93:GLA-SE when compared with ID93 alone, with higher production of polyfunctional T-cell expressing TNF-α and IL-2 but not IFN-γ which is consistent with preclinical studies. A higher frequency of antigen specific antibodies were produced by animals vaccinated with ID93:GLA-SE, namely belonging to IgG1 and IgG3 subclasses, suggesting that GLA can increase antibody production [36]. The latter could be important since previous data suggests that this type of response can help to decrease extra-pulmonary dissemination of Mtb and increase phagocytosis [36].

Taking in consideration previous publications and clinical trial NCT01927159, researchers decided to investigate the potential of ID93:GLA-SE to be used as a therapeutic vaccine in combination with front-line antibiotics in a challenged mice model [37]. The results were analyzed by comparing the vaccine-antibiotic group versus the antibiotic only group and the results showed an increase survival, an enhanced bacterial load control and a decrease in lung pathology in the vaccine-antibiotic group. This suggests that ID93:GLA-SE is a candidate for therapeutic vaccination in combination with antibiotics [37]. Immune responses were Th1-biased and, accordingly, in the vaccinated group the T-cells were polyfunctional and released IL-2, IFN-γ and TNF-α [37], while in the non-vaccinated group the response presented a Th2-biased pro-inflammatory profile [37].

One of the strong points of the research using this adjuvant, GLA-SE, is the study of its immunogenic properties, as well as the protection given by the vaccine candidate ID93:GLA-SE, against clinical isolates and virulent strains. In fact, most of the mice studies for TB vaccines are performed using challenges with laboratory strains that are not circulating in humans in over 70 years, which means that results of preclinical studies may not be fully reliable in pointing out the real potential of the vaccine [34]. The ID93:GLA-SE has been tested for its protection against isolates from the Beijing family, which are related to multi-drug resistance and are associated with 13% of TB outbreaks in the world, although they are particularly prevalent in East Asia [35,36,40]. The studies that tested ID93:GLA-SE against strains from the Beijing family in BCG non-vaccinated mice showed that the vaccine candidate was able to meet the outcomes of the studies, namely decreased bacterial burden and reduced lung pathology [34,35]. In another study the vaccine increased survival and protection against pulmonary TB and induced higher lymphocytes population in the lung infiltrates than in the control and placebo groups [34]. Mice in both studies had similar immune profiles, characterized by a long-lasting Th1 type response with the production of polyfunctional T-cells producing IFN-γ, IL-2 and TNF-α [34,35]. In one study, CD8+ effector cells were detected in the vaccinated mice [35] while in the other no CD8+ cells were detected [34] which is one of the indicators that further studies are needed to validate this strategy. As a BCG booster in mice challenged with a Beijing strain, ID93:GLA-SE also showed encouraging outcomes such as a decrease in bacterial load and in lung damage, characterized by regular shape granulomas that contrasted with the irregular and large granulomas in the non-vaccinated or BCG only groups, accompanied by a Th1 type of immune response similar to the one described above [39]. By showing the potential of this vaccine candidate to work as a BCG booster, this study provides promising results since a major part of the population is vaccinated with BCG and this vaccine is still effective in the pediatric population. Currently, there are two active clinical trials in South Korea that were designed based on these results (NCT03806699 and NCT038006686).

#### 4.1.3. AS01

AS01 is an already licensed adjuvant system used in the Herpes vaccine Shingrix^®^ and it is a formulation with two compounds with adjuvant properties. The first is MPL, a TLR4 agonist previously named GLA, and the second is the natural saponin QS-21 [40]. This adjuvant system has shown to be able to induce antibody and T-cell responses when combined to different antigens, both in preclinical as in clinical phase, due to a synergistic effect of its components that are formulated into a cholesterol liposome [40,41]. MPL is a well-established TLR4 agonist derived from the chemical transformation, and thus detoxification of *E. coli* lipid A. It stimulates TLR4-mediated cytokines production through both MyD88 and TRAM/TRIF pathways, while saponin QS-21 induces the production of IL-2, IFN-γ, and IgG2a antibodies, which are characteristic of a Th1 response. This saponin has also been described as a inducer of NLRP3 inflammasome, although this is not yet fully understood [41]. AS01 is thought to induce a protective cellular response by starting several innate immunity pathways. Upon injection, site inflammation, and lymph node drainage, AS01 stimulates the activation of efficient Antigen Presenting Cells (APC) [42].

In the TB vaccine candidate M72:AS01_E,_ the designation AS01_E_ is used, since it refers to half dose of the adjuvants in the formulation [40]. M72 is a recombinant fusion protein of two Mtb antigens, Mtb32A and Mtb39A [43]. This vaccine has been extensively studied and several clinical trials have been developed to test its safety, immunogenicity, and efficacy in different cohorts, after intramuscular administration [44,45].

The phase II trials were performed on different patients’ cohorts to evaluate the vaccine’s ability to protect against TB in different stages of the disease and in different ages and health conditions. A phase IIb trial targeting HIV-negative participants with latent TB infection was performed in endemic countries (NCT0175598) under the premise that a vaccine capable of preventing pulmonary TB in already infected patients could stop disease progression, since Mtb is not transmitted in the latent phase of the disease [46,47]. Most of the participants were also vaccinated with BCG. One of the limitations attributed to the study was the failure to detect early active cases of pulmonary TB that could have altered the efficacy of the vaccine overall [46,47]. Curiously, another trial attempted to include participants with active TB, but in later stages of recovery, to assess if the M72:AS01_E_ was effective in preventing TB in patients that were border between latent and active disease [48]. However, due to high reactogenicity in this group of participants, the trial recruitment was terminated and the authors hypothesized that this outcome was due to the effect of the AS01 immunogenicity in the patients with active disease [48]. Nevertheless, in the trials that included this vaccine candidate it was possible to characterize the immune response. Accordingly, it was described that this vaccine could increase antigen specific CD4+ cells and that the response was Th1 type, with the identification of characteristic cytokines such as IFN-γ, TNF-α and IL-2 as well as polyfunctional cells that expressed all of the three or combinations of the three cytokines [47,48]. In the adult, HIV negative and Mtb infected efficacy trial, the vaccine’s efficacy after a three-year follow-up period was 49.7% [47], which is approximately the expected 50% that the WHO described as desired [15]. Although this is promising, extensive testing is required before advancing in the clinical trial process. Interestingly, there is a sub-study trial (NCT02097095) that aims to further characterize the immune profile by extensively analyzing the samples from the participants, in the hope to establish immune correlates for TB protection [47].

Currently, there is an active clinical trial (NCT04556981) that aims to evaluate the safety and immunogenicity of M72:AS01_E_ in HIV-positive patients with controlled infection. In a previous trial (NCT01262976) that evaluated the same parameters in HIV positive and HIV negative participants, the vaccine was found to be safe and able to induce antigen-specific antibodies as well as polyfunctional CD4+ T-cells in all of the groups, which was a very positive finding [49]. This proved that HIV positive patients can have an immune response, at least those who are not severely immunocompromised which was the case of these participants, to this vaccine and that the vaccine can possibly meet another requirement for WHO preferred characteristics [15].

#### 4.1.4. CAF01

Cationic Adjuvant Formulation 01 (CAF01) is a liposome formulation composed by N,N-dimethyl-*N,N*-dioctadecylammonium (DDA) and α,α-trehalose 6,6′-dibehenate (TDB) [50]. This adjuvant system was projected to explore the properties of DDA in a stable formulation, although TDB is also able to potentiate the immune responses [50]. This adjuvant has been described as promoter of Th1 and Th17 responses and its ability to induce long living memory responses is also documented. One possible mechanism of action is the activation of the C-type lectin MINCLE receptor by TDB that leads to NF-KB signalling [50]. Experimentally, it has been described, by radiolabelling, that this adjuvant is able to form a depot at the injection site and adsorb the antigen for up to six days, which is also part of its mechanism of action [51].

There are a few recent studies that explore the potential of CAF01 as a TB vaccine adjuvant, mainly due to the Th1/Th17 bias response that it is able to trigger [51,52,53,54,55]. In all of these investigations, the vaccine candidate is a combination of CAF01 with the antigen H56, which is actually a fusion protein of 3 Mtb antigens—Ag85b, ESAT-6 and Rv2660 [47,48,49,50,51]. Woodworth and co-workers, performed an in vivo study that provided insight on the immune responses triggered by this vaccine candidate [53]. In vaccinated animals, the bacterial burden was lower, and T-cell response was present before infection while on the control groups, T-cell response was caused by Mtb priming and only appeared after the onset of infection [53]. Additionally, they concluded, using an in vivo labelling technique, that IL-2 and IL-17 producing T-cells, with a KLRG1- CXCR3+ phenotype, were present in the lung parenchyma of the vaccinated mice and, when in circulation, were able to migrate to the lung upon infection with Mtb [53]. With this investigation they provided the possibility that Th17 type of immunity is also needed for protection against infection and that it can be achieved with subcutaneous administration and not only with mucosal delivery [53]. In a following investigation, the same authors hypothesised if an intranasal administration of the H56:CAF01 vaccine would boost the immunization provided by the first two subcutaneous injections [52]. The findings further confirmed the cellular response of the previous study but showed that the increase in T-cell lung population elicited by the mucosal booster did not result in increased protection against infection or differences in Th1/Th17 profile, when compared to subcutaneous administration only [52]. Similarly, another study compared the homologous boosting of H56:CAF01 using a intrapulmonary administration after a previous parental priming with three subcutaneous injections [51]. In this case, the authors did not challenged the mice with Mtb and measured the immune response, namely the Th1 and Th17 associated cytokines and produced antibodies [51]. The authors observed that, in the respiratory tract, the promotion of Th17 and Th1 responses, including the production of polyfunctional cells as well as cells with the ability to migrate into the lung parenchyma, were strong. Furthermore, a CAF01 dose-dependent relationship was established for induction of systemic Th1 and Th17 cells [51]. A strong IgA response was also observed, although the association of humoral responses with Mtb protection was not very clear, despite the fact that some researchers are attempting to define the contribution of humoral immunity to protection and pathogenesis [51,56]. Nevertheless, in terms of mucosal immunity, IgA appears to have an important role [51]. While preliminary, these studies show that it might be promising to pursue a mucosal immunization strategy, upon optimization of the vaccine’s formulation. Appropriately, some researchers are studying formulations for intrapulmonary administration by inhalation [54,55]. For example, Thakur and co-workers described the ability of a spray-dried formulation to elicit Th and Th17 comparable to subcutaneous administration [54]. The boosting strategy is being pursued by researchers however, not only the use of a different administration route was explored. A boosting strategy with the antigen H56 alone, after priming with the H56:CAF01 vaccine candidate showed, using transcriptomic approaches, the immunostimulatory effect of the adjuvant, by comparing with the control groups [57]. By priming the mice with a H56:CAF01 vaccine, the response obtained with the subsequent administration of the antigen alone revealed a stronger increase in immune response genes transcription [57]. These findings can be a starting point to further explore the innate cells role in TB protection as well as their ability to create memory [57]. While the potential of this vaccine candidate seems promising, at the moment this review was written (Dec 2020) there are no active clinical trials with it. Table 2 summarizes the adjuvant systems that have been described in this section.

### 4.2. Adjuvants in TB Vaccines Currently in Preclinical Studies

#### 4.2.1. Starch

Raw starch microparticles are an interesting adjuvant to be used in vaccines against TB. The starch molecule itself, which is basically a carbohydrate, is safe, biodegradable and biocompatible [58]. Structurally, starch is an α-glucan and this characteristic is important to explain a possible mechanism of action [59]. This structure, alongside the particles size and format, has a resemblance to one of the constituents of the cell wall of Mtb, a glycogen-like α -glucan composed of α-(1→4)-D-Glc core with mono and diglucoside, every five to six residues [59]. When administrated intranasally, the starch microparticles are therefore able to mimic the interaction along the respiratory tract where they are recognized by α-glucan specific receptors, namely C-type lectin DC specific ICAM-3-grabbing nonintegrin, and begin to elicit a specific immune response [59]. Moreover, some previous reports proposed that TLR6 signalling is involved and a Th1 response is triggered [60,61] although new and more detailed studies are lacking and, additionally, immunogenicity due to the size and charge is also observed since these properties make starch microparticles more susceptible to phagocytosis [59]. Besides its interaction with receptors, starch microparticles have also been described as an excellent delivery system for the chosen antigen [58,62]. The formulations containing this adjuvant are administrated intranasally and pursue a strategy towards the development of a mucosal vaccine for TB [59].

An interesting in vivo study hypothesized the use of starch microparticles as an adjuvant for a heat-shock protein antigen vaccine or alone as a BCG booster and showed promising results as the bacterial load was decreased in the animals that received the booster [59]. Moreover, the starch adjuvant properties were confirmed by the study design’s outcomes since no significant difference between the boosters was observed and both contained starch microparticles [59]. In a subsequent trial, the same authors explored the ability of starch microparticles to be adjuvants to the BCG vaccine [63]. The authors evaluated overall mice survival and bacterial load and the results showed that the groups that were vaccinated with BCG and the adjuvant had a higher survival and a lower bacterial load, although the damages to the lung were not different from the BCG only vaccination group [63]. In these experiments, the authors decided to use clinical isolates and also a challenge with a higher concentration of particles of Mtb. While it is interesting from a research point of view, since it aims to be more predictive of reality, it clearly shows that this type of studies is not standardized. Furthermore, no immune parameters were explored in detail, although authors hypothesized, based on previous findings, that starch microparticles could promote CD4+ and CD8+ responses and Th1 response [63].

#### 4.2.2. Chitosan

Chitosan is a polysaccharide adjuvant that has been explored as a potential adjuvant for vaccines against TB in the last few years [64,65,66,67]. Similarly to starch, chitosan has several advantages related to its structure and origin, namely low toxicity, biocompatibility and biodegradability [68]. Structurally, chitosan is a linear polysaccharide of β-(1→4)-_D_-glucosamine and N-acetyl-_D_-glucosamine, randomly distributed [65]. Moreover, chitosan formulations also exhibit appropriate size and charge that favour mucoadhesivity, ability to penetrate between cells, improved cell uptake, controlled release of the antigen, and improvement in its presentation to specialized cells [65,68]. It has been described that chitosan is immunogenic by activating the inflammasome and leading to the release of pro-inflammatory cytokines, namely IL-1β and IL-18, that ultimately lead to the activation of Th1, Th2 and Th17 responses [65].

A recent study about the use of chitosan-based nanoparticles as adjuvant and delivery system shows how this type of adjuvants can be incorporated into a more complex formulation. Poecheim and co-workers developed and characterized a trimethyl chitosan (TMC) nanoparticle delivery system to incorporate a plasmid DNA encoding for a Mtb antigen and also a muramyl dipeptide [66]. The work, that included in vitro and in vivo studies, highlighted different advantages of this vaccine candidate, namely the ability to promote Th1 type of response, which was evaluated trough the production of IFN-γ by T-cells in the spleen and also by the IgG2c/IgG1 ratio [66]. Since IgG2c is associated with a Th1 response, when this ratio is higher than 1, it is considered a Th1 biased response [66]. The interesting aspect of this vaccine candidate is the combination of different adjuvants in one formulation, besides TMC, the plasmid DNA containing CpG motif that stimulates TLR9 and the muramyl dipeptide that activates NOD-like receptor 2 and promotes Th1 responses [66]. These responses are all enhanced by the ability of the TMC nanoparticles to deliver this components inside the cell in a phagocytosis-dependent mechanism and cause dendritic cell maturation [66]. All of these mechanisms synergically stimulate a Th1 response [66]. In this in vivo study, the mice were immunized using a intramuscular injection [66].

Another benefit from the use of chitosan as an adjuvant, is the possibility to explore different administration routes, namely intranasal, that can be advantageous when it comes to vaccines for respiratory pathogens like Mtb [69]. Amini and co-workers developed a TMC:ESAT-6 vaccine candidate to be administrated intranasally [67]. The logic behind this formulation is that TMC nanoparticles can encapsulate the antigen and slowly release it, increasing the uptake by immune cells in the nasal-associated lymphoid tissue, which starts a specific immune response in the respiratory mucosa [67]. The measured parameters in this case were the production of IFN-γ, antigen-specific antibodies, and IL-4 [67]. As in the other study previously presented in this paragraph, although the outcomes show that the vaccine candidate was active in stimulating immunity [67], it would be interesting to further characterize the mechanism of action, also investigating other cellular and humoral responses caused by this adjuvant/delivery system.

Yu and co-workers designed a study of a subcutaneous vaccine candidate that was able to provide more information about immune responses associated with chitosan. The adjuvant consisted in inulin and chitosan chemically conjugated and the antigen a fusion protein of two Mtb antigens, CFP10 and TB10.4 [64]. This in vivo study on mice, showed that the formulation was able to induce both a Th1 and Th2 response, which was confirmed by the release of IFN-γ, TNF-α, IL-2, as well as IL-4, and the release of antigen-specific antibodies [64]. The ability to increase the fusion’s protein exposure to immune cells was also confirmed and it was demonstrated that the vaccine with the adjuvant had a decrease in renal clearance and in proteolytic digestion [64]. Once again, it was demonstrated that chitosan can be combined with other adjuvants to improve the vaccine’s pharmacokinetic and pharmacodynamic properties. While positive effects prevail, chitosan has some drawbacks as the difficult removal from circulation due to its high polarity and positive charge and the possibility of tolerance induction [65]. These issues should be addressed if any of the following vaccine candidates achieve access to clinical trials.

#### 4.2.3. Other Adjuvants in Preclinical Studies–Cyclic Dinucleotides and Advax^®^ Formulations

The adjuvants mentioned above are just part of a bigger pipeline that is constant development and extension. In fact, there are some other new adjuvants that not only aim to be effective in starting a protective immune response against TB, but also give new insights about the protective immune responses and possible correlates and biomarkers of protection. A summary of the described adjuvants in preclinical development is described in Table 3.

Cyclic dinucleotides (CDNs) are being studied as possible adjuvants [70]. These molecules are actually pathogen-associated molecular patterns (PAMPs) and activate the cytosolic receptor stimulator of interferon genes (STING) which in turn begins a signalling trough different pathways [70]. A vaccine candidate formulated with CDNs and the commercially available squalene oil-in-water nano-emulsion adjuvant AddaVax^®^ was tested in mice and, interestingly, different administration routes elicit different cellular responses, specifically, subcutaneous administration results in Th1/Th2 responses and mucosal administration is associated with Th17, and the latter was associated with enhanced immune response and enhanced protection against TB on mice [70].

Advax^®^ is a delta-inulin microparticles adjuvant with a positive safety profile, which has shown to be able to enhance T and B cells activation and promote a Th1 and Th2 type of response due to its ability to increase phagocytosis and recruit cells [68,71]. An Advax^®^ and CpG adjuvant system was used in a TB vaccine candidate, CysVac2 [71]. After intramuscular immunization with this formulation, a high frequency of polyfunctional T-cells that expressed IFN-γ, TNF-α, and IL-2 concomitantly was observed, confirming a Th1 type of response [71]. Furthermore, after challenge with a Mtb aerosol, the vaccinated mice showed increase protection when compared to placebo, as well a decrease in bacterial load and lung pathology [71]. Advax^®^ was also formulated with CpG and a muramyl dipeptide, murabutide, in a vaccine candidate with recombinant fusion proteins. This adjuvant system combines, besides the inulin, a TLR9 agonist, which is the CpG and a NOD-like receptor 2 agonist, the murabutide peptide [72]. In this study, the mice immunized intramuscularly with the combination of fusion protein and adjuvant system showed antigen-specific antibodies, namely IgG2a and IgG1, at higher levels than the other groups, with a bias to IgG2a which suggests a Th1 type of response [72]. Moreover, the mice in this group also had decreased lung inflammation after challenge with Mtb [72]. Both of these studies show the potential of Advax^®^ and possibly other inulin-based adjuvants in TB vaccine formulations.

## 5. Future Perspectives and Conclusion

New TB vaccines are needed, and it is clear that the adjuvants in the formulations can play a determinant role in the success or failure of the vaccine as they help to modulate the immune response and optimize the antigen’s presentation through different administration routes.

Due to the natural infection route, mucosal administration has been actively pursued by many researchers since it has the potential to provide physiological and immunological advantages against a Mtb infection, although there are safety concerns associated with adjuvants and this administration that must be taken into account [12,73]. Moreover, the exploration of Th17 responses is quoted by some authors as a new perspective for TB vaccines, since some vaccine candidates selectively inducing just Th1 response failed during the years [74].

As illustrated by some of the described examples, synergistic effect between two or more adjuvants might be further explored in the future, which in turn can stimulate different innate immune pathways (TLRs, NOD receptors, CTL receptors). The formulation of adjuvant systems that take advantage of two or more adjuvants in the vaccine candidate has already been introduced in the TB vaccine pipeline, that is the case of AS01 and IC31, but others have been developed. For example, the novel adjuvant MTOM consists in MPL, trehalose-6.6′-dibehanate, MF59, and heat-killed *Mycobacterium vaccae* and has shown the ability to enhance Th1-type response [75].

Ideally, the future of TB vaccine development should include the establishment of correlates of protection and biomarkers that can be studied in every preclinical and clinical investigation [17]. The development and standardization of new techniques capable of extensively characterizing a formulation and predict its immunogenicity could give precious data before testing on animals and this could help decrease the costs of TB vaccine research and minimize the use of animal models [26]. Furthermore, it could also provide information about the safety, immunogenicity, and physicochemical properties of adjuvants which in turn would contribute to the production of safer new vaccine candidates [41].

This review aimed to give insights on adjuvants used in preclinical and clinical studies in the search for a new effective TB vaccine published in the last five years. It is clear that the general trend is the development of adjuvants capable of starting a strong and durable Th1 response, with the production of polyfunctional cells, however we also show important examples of adjuvant formulations that focus on other responses, such as Th17, which is very important for intracellular pathogens like Mtb. Antibody humoral response is being neglected in the majority of studies reviewed here, but in our opinion, it should be reconsidered for a more global evaluation of the outcome of the vaccine.

## Figures and Tables

**Figure 1 cells-10-00078-f001:**
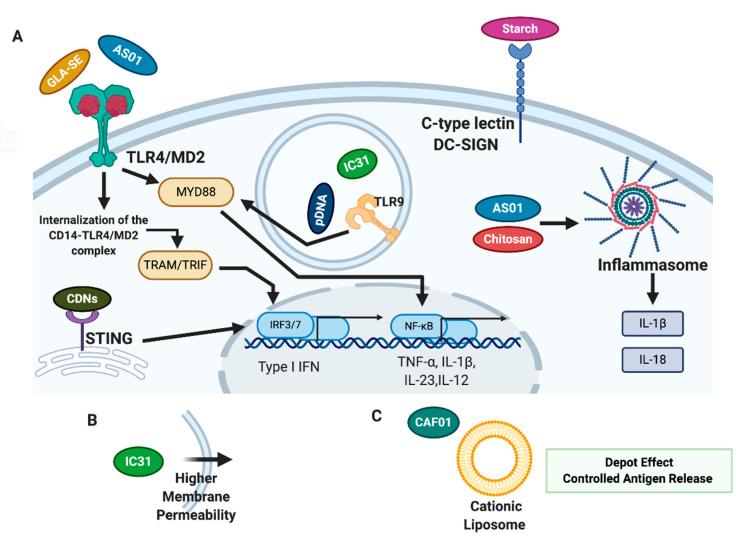
O verview of Proposed Mechanisms of Action of Adjuvants in vaccines. (**A**) Receptors and Molecule Activation of adjuvants. The active principle of GLA-SE and AS0, the MPLA, activates TLR4 and downstream pathways, namely MyD88 and, after internalization of the CD14/TLR4/MD-2 complex, the “late” TRAM-TRIF pathway. Starch interacts with C-type DC-SIGN receptor. IC31, more specifically, CpG motifs, as well as pDNA present in TMC microparticles activate TLR9 inside the endosome. AS01 and Chitosan activate the inflammasome and promote the subsequent release of pro-inflammatory cytokines. CDNs activate STING receptors in the endoplasmic reticulum. (**B**) IC31′s KLK peptide increases the permeability of the adjuvant system in the membrane to deliver the antigen and CpG motifs. (**C**) Cationic Liposome adjuvants, such as CAF01, promote a depot effect on the injection site and release the antigen in a controlled way, stimulating innate immunity.

**Table 1 cells-10-00078-t001:** Summary of a new vaccine’s WHO preferred product characteristics [15].

**Target Population**	Adolescents and adults
**Outcome Measure and Efficacy**	50% or greater efficacy in preventing confirmed pulmonary TB
**Duration of protection**	Ten years or more
**Safety**	Favourable safety profile, even for high-risk groups as HIV patients
**Schedule**	Less than three doses to achieve primary immunization and booster preferentially after 10 years or more
**Co-administration**	Safe and without interactions with other vaccines administrated to the same population
**Immunogenicity**	Characterization of immune markers and concomitant development of correlate of protection of a TB vaccine
**Programmatic Suitability and Prequalification**	Should meet requirements of WHO suitability of vaccines—vaccine presentation, packaging, thermostability, formulation and disposal
**Value Proposition**	Favourable cost-effectiveness and affordable price

**Table 2 cells-10-00078-t002:** Summary of Adjuvants in TB vaccines Currently in Clinical Stage of Development.

Adjuvant System	Components	Proposed Mechanism of Action	Type of Immune Response	Vaccine Candidate	Immunization Strategy	Adm. Route	Ref
IC31	KLK, ODN1a	TLR9 activation (ODN1a) Enhanced delivery of ODN1a to the endosome, enhanced antigen presentation (KLK)	Th1–Polyfunctional T-cells producing IFN-γ, IL-2 and TNF-α	H4:IC31 H56:IC31	Prophylactic Prophylactic, Post-Exposure	I.M. I.M.	[22,23,26,27,28,29,30]
GLA-SE	GLA in a Squalene oil-in-water emulsion	TLR4 activation	Th1–Polyfunctional T-cells producing IFN-γ, IL-2 and TNF-α Antigen-specific IgG1 and IgG3 production	ID93: GLA-SE	Prophylactic, BCG booster, Therapeutic	I.M.	[32,33,34,35,36,37,38,39]
AS01	MPL, QS-21	TLR4 activation (MPL) Induction of NLRP3 inflammasome (QS-21)	Th1–Polyfunctional T-cells producing IFN-γ, IL-2 and TNF-α	M72:AS01_E_	Post-exposure, BCG booster	I.M.	[42,43,44,45,46,47,48,49]
CAF01	DDA, TDB	MINCLE activation Depot Effect Controlled release of the antigen	Th17–T-cells expressing IL-17 Th1 IgA response	H56:CAF01	Prophylactic Homologous Boosting	S.C. I.N.	[50,51,52,53,54,55,57]

ODN1a–oligodeoxynucleotide (ODN) 1a; TLR–Toll-Like Receptor; MPL–3-O-desacyl-4′-monophosphoryl lipid A; BCG–Mycobacterium bovis bacilli Calmette-Guérin; CAF01–Cationic Adjuvant Formulation 01; DDA–N,N-dimethyl-N,N-dioctadecylammonium; TDB–α,α-trehalose 6,6′-dibehenate; I.M.–Intramuscular; S.C.–Subcutaneous; I.N.-Intranasal.

**Table 3 cells-10-00078-t003:** Summary of Adjuvants in TB vaccines Currently in Preclinical Stage of Development.

Adjuvant System	Components	Proposed Mechanism of Action	Type of Immune Response	Adm. Route	Ref
Starch Microparticles		C-type lectin DC specific ICAM-3-grabbing nonintegrin receptor activation Increase in phagocytosis and macrophages activation TLR6 signaling	Th1	I.N.	[58,59,60,61,62,63]
Chitosan		Inflammasome activation Mucoadhesive, ability to penetrate between cells, controlled release of the antigen, improved cell uptake	Th1 –IFN-γ production, IgG2c Th2 Th17	I.M.	[65,68]
TMC nanoparticles	TMC	DC maturation Increase in antigen’s intranasal residence Increase in the antigen’s uptake	Th1 Th2 Antigen-specific antibody production	I.N. (TMC-ESAT-6)	[68]
TMC Plasmid DNA, Muramyl peptide		DC maturation TLR9 activation (Plasmid DNA) NOD-like receptor 2 activation (muramyl peptide)	Th1 –IFN-γ production, IgG2c	I.M.	[67]
Chitosan-Inulin	Chitosan Inulin	Increase antigen’s exposure to immune cells. Decrease in renal clearance and in proteolytic digestion	Th1- Polyfunctional T-cells producing IFN-γ, IL-2 and TNF-α Th2–T-cells producing IL4 Antigen-specific antibodies–IgG1 and IgG2b	S.C.	[64]
CDN-AddaVax^®^	CDNs Addavax^®^ (oil-in-water emulsion)	STING activation. (CDNs) Enhanced T-cell and B-cell activation (AddaVax^®^)	Th17 Th1 Th2	S.C. I.N.	[70]
Advax^®^-CpG	Delta-inulin micropaticles (Advax^®^) CpG	Enhanced phagocytosis and cell recruitment. (AddaVax^®^) Enhanced T and B cell activation. (AddaVax^®^) TLR9 activation (CpG)	Th1–Polyfunctional T-cells producing IFN-γ, IL-2 and TNF-α	I.M. (CysVac2)	[71]
Advax^®^-CpG- murabutide	Delta-inulin micropaticles (Advax^®^) CpG Muramyl dipetide (murabutide)	Enhanced phagocytosis and cell recruitment. (AddaVax^®^) T and B cell activation. (AddaVax^®^) TLR9 activation (CpG) NOD-like receptor 2 activation (muramyl peptide)	Th1–IgG2a and IgG1 production with a IgG2a bias.	I.M.	[72]

TLR–Toll-Like Receptor; TMC–Trimethyl Chitosan; DC–Dendritic Cell; CDNs–Cyclic Dinucleotides; STING–Stimulator of interferon genes; I.M.–Intramuscular; S.C.–Subcutaneous; I.N.–Intranasal.

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
