# Peer review of "Developing New Anti-Tuberculosis Vaccines: Focus on Adjuvants"

_cells, 2021, doi:10.3390/cells10010078_

Round 1
Reviewer 1 Report
Very good work summarizing the work that has bene already published in an evolving area. Text reads well without overwhelming details
Areas of improvements
-The STING agonist in TB is included in the figure one but then is being described briefly in the conclusion/ future direction part. It should be moved in the preclinical studies part (Cell reports; 2017)
-Addavax is noted for first time in conclusion too; would move it; has been also used in preclinical studies and combined with STING agonist successfully (Cell reports; 2017)
- Would propose clear report of the route of administration of each adjuvant and also if co-formulated with the vaccine of interest and would clarify if the reported immunogenicity results each time was after the concurrent use of electroporation
-I would propose a consistent structure of the paragraph/section per adjuvant. Would propose this paragraph/section to include consistently
-molecular background
-possible mechanism of action
-route of administration
-discuss available data in clinical and preclinical studies and report the results of each study explicitly including both immunogenicity and challenge/protection studies. Clarifying each time what is known and what is pending
-discuss consistently Side effects/ safety concerns
-Consider more tables representing basic conclusions/ established knowledge per adjuvant
-The conclusion section should not add more new info with adjuvants that have not been mentioned before in the main body of the review. It should summarize and indicate more clearly what is established and what is not.
Author Response
Please find point-by-point answers (bold) to reviewer's comments:
-The STING agonist in TB is included in the figure one but then is being described briefly in the conclusion/ future direction part. It should be moved in the preclinical studies part (Cell reports; 2017)
Thank you for pointing this out. We agree with the suggestion and we have added a new topic in the body of the article with these adjuvants and included them in the preclinical studies part. Taking this comment in consideration, we have also moved the Advax adjuvant to preclinical studies.
-Addavax is noted for first time in conclusion too; would move it; has been also used in preclinical studies and combined with STING agonist successfully (Cell reports; 2017)
We appreciate this suggestion. In this review we have not explored Addavax in detail due to the lack of recent articles in its use in tuberculosis vaccine. But, nevertheless, we have highlighted its association with STING agonists and recognize that not to mention this was a shortcoming of the article before.
- Would propose clear report of the route of administration of each adjuvant and also if co-formulated with the vaccine of interest and would clarify if the reported immunogenicity results each time was after the concurrent use of electroporation
We thank the referee for this useful suggestion. We added the administration route of the vaccine candidates described as we agree that it is an important part of our review. This is especially true since we explore the possible use of a mucosal route as a promising TB vaccine approach. We do not believe that exploring the use of electroporation is within the scope of this particular review.
-I would propose a consistent structure of the paragraph/section per adjuvant. Would propose this paragraph/section to include consistently
-molecular background
-possible mechanism of action
-route of administration
-discuss available data in clinical and preclinical studies and report the results of each study explicitly including both immunogenicity and challenge/protection studies. Clarifying each time what is known and what is pending
-discuss consistently Side effects/ safety concerns
We have taken in consideration reviewer’s suggestion and have made some changes throughout the text to try to organize better the structure of the paragraph, according to reviewer’s scheme. Overall, we believe that this contributed to improve the clarity of the review.
-Consider more tables representing basic conclusions/ established knowledge per adjuvant
We thank you for this recommendation. We have added two tables, one for adjuvants in clinical stages and another one for adjuvants in preclinical studies, that aim to summarize these key aspects. We hope that this makes the review more comprehensive and easier to read.
-The conclusion section should not add more new info with adjuvants that have not been mentioned before in the main body of the review. It should summarize and indicate more clearly what is established and what is not.
Thank you for this comment. In fact, we agree that we should not add new information about adjuvants in the conclusion section. We have revised this issue and moved the new information to the preclinical studies section. We left an information about a new adjuvant system in the future perspectives to highlight this consisting direction but, in this case, it is a system composed of commercially available adjuvants and not a new molecule and we aim just to give a short insight on what the future can be.
Reviewer 2 Report
Dear Authors,
the manuscript entitled 'Developing New Anti-Tuberculosis Vaccines: Focus 2 on Adjuvants' is an exaustive and comprehensive report on recent developments about vaccines against tuberculosis. This review rely on the most recent tubercolosis vaccines developments and it displayed the cell mechanisms at basis of the immunogenic reactions. The review is complete, well written and organized. For all these reasons I suggest its pubblication on Cells.
PS: The authors should better check the form of the references paragraph
Author Response
Dear Reviewer,
Thank you very much for your appreciation of the review.
According to your suggestion we have updated the reference paragraph to match the template from Cells.
With our best regards
Reviewer 3 Report
This is a well conceived review about the adjuvants being developed for TB vaccines though with some shortcomings.
Although, it covers only selected few adjuvants, it provides an overall scientific rationale and their promising activity as adjuvants.
- At places it lacks the information about the clear mechanism of action of different adjuvants discussed in this review.
- A more specific discussion about the adjuvants that could be more suitable for subunit vaccines and attenuated whole organism vaccine should be added.
- Some more recently identified adjuvants targeting intracellular pattern recognition receptors could be included in this review to make it more comprehensive.
- Manuscript also needs a thorough English language editing to improve its quality for native English readers.
Author Response
Dear Reviewer,
please find point-by-point answers (in bold) to your comments.
With our best regards
At places it lacks the information about the clear mechanism of action of different adjuvants discussed in this review.
We have taken this comment in consideration and attempted to improve the description of the mechanism of action in the adjuvants that were not so clear. For example, in the case of the CAF01 adjuvant, the role of TDB was further characterized and explain in a summarized way.
A more specific discussion about the adjuvants that could be more suitable for subunit vaccines and attenuated whole organism vaccine should be added.
We appreciate this comment and the suggestion to add this distinction between adjuvants that could be more suitable for subunit vaccines and whole organism, however, in our opinion this discussion falls outside the scope of this review that is focused on subunit vaccines against TB.
Some more recently identified adjuvants targeting intracellular pattern recognition receptors could be included in this review to make it more comprehensive.
As from previous point, review’s focus is the development of adjuvants for TB vaccines, in this context, and at the best of our knowledge, all types of adjuvants also targeting PRRs (in particular TLRs) have been considered.
Manuscript also needs a thorough English language editing to improve its quality for native English readers.
Thank you for this observation. We have improved English language and eliminated some typos